# Does microcredit for the poverty-alleviated population have an income-increasing effect? Also on the "Raising the Low" effect

Yin Liu[1,2‡], Lu Fan [3‡], Binjian Yan [1*]

1 School of Economics and Management, Nanjing Agricultural University, Nanjing, Jiangsu, China,
2 School of Public Administration, Xinjiang Agricultural University, Ürümqi, Xinjiang, China, 3 School of Urban Economics and Public Administration, Capital University of Economics and Business, Beijing, China

‡ YL and LF also contributed equally to this work.
* 15129367326@163.com

## Abstract

Promoting the continuous income increase of the poverty-alleviated population has become one of the key tasks to consolidate the achievements of poverty alleviation, promote rural revitalization and achieve common prosperity. Based on the survey data of 738 farmers in six poverty-alleviated counties (cities) in an underdeveloped region in northwest China, this paper uses the endogenous transformation regression model to empirically explore the income-increasing effect of microcredit for the poverty-alleviated population, and uses the quantile regression model and OLS regression model to explore raising the low effect of microcredit for the poverty-alleviated population. The results show that: (1) Based on the counterfactual hypothesis, if the farmers who obtain microcredit for poverty-alleviated population are not loaned, their production and operating income will decrease by 3.31%, that is, microcredit has a significant income-increasing effect, and the income-increasing effect of obtaining microcredit for the poverty-alleviated population on the monitoring objects is greater than the households that have lifted out of poverty. (2) The mechanism of action shows that the microcredit policy for the poverty- alleviated population promotes income increase by promoting farmers to increase material capital investment and social capital investment. (3) The income-increasing effect of obtain microcredit for poverty-alleviated population on the low-income initial endowment farmers is greater than that on high-income initial endowment farmers, and farmers with low-land initial endowment have a higher income increase effect, that is, a 'raising the low ' effect. The overall efficiency effect on 'low→high 'initial endowment farmers shows a decreasing trend. Therefore, in order to ensure the effectiveness of financial precision assistance, we should promote the microcredit policy of the poverty-alleviated population from the aspects of policy stability and implementation precision.

**Data availability statement:** All relevant data are within the manuscript and its Supporting Information files.

**Funding:** The author(s) received no specific funding for this work.

**Competing interests:** The authors have declared that no competing interests exist.

# 1. Introduction

In 2020, the decisive victory of poverty alleviation has been achieved. China has historically eliminated absolute poverty, opened a new journey of rural revitalization, and entered a new historical stage of solidly promoting common prosperity. To promote common prosperity, the most arduous and onerous task is still in the countryside. To realize the common prosperity of farmers in rural areas is a necessary condition and proper meaning for China to realize the common prosperity of all the people. Consolidating the achievements of poverty alleviation and preventing large-scale return to poverty are the bottom line requirements for realizing the common prosperity of farmers in rural areas, and promoting the faster growth of income of low-income farmers is the core task of realizing the common prosperity of farmers in rural areas [1]. At the present stage, the monitoring and assistance of low-income people to promote their continuous income increase has become one of the key tasks to consolidate the achievements of poverty alleviation, promote rural revitalization and achieve common prosperity.

At the end of 2024, the Central Rural Work Conference emphasized the prevention of poverty return monitoring, the implementation of assistance measures, the enhancement of endogenous motivation, and the continuous consolidation and expansion of poverty alleviation achievements, and the resolute adherence to the bottom line of no large-scale poverty return. As a representative financial support measure led by the government, during the period of comprehensively promoting rural revitalization, microcredit for poverty-alleviated population helps poverty-alleviated households and monitoring households to carry out production and development operations, and has become an important engine to enhance the endogenous development momentum of poverty-alleviated population. Continue to further promote the microcredit of poverty-alleviated population, maintain the intensity of credit delivery in poverty-alleviated areas, and fully implement the support policy of microcredit for the poverty-alleviated population to "provide loans to all who need them", which has become an important starting point to promote the effective connection between the achievements of poverty alleviation and rural revitalization, and to achieve common prosperity. Based on this, Studying whether the microcredit for the poverty-alleviated population has an income-increasing effect, how effective is its implementation, and what is the mechanism of action? Can we achieve the goal of 'raising the low' and 'expanding the middle'? It is of great practical significance in promoting rural revitalization and achieving common prosperity.

# 2. literature review

Credit constraint is one of the main obstacles to the development of rural inclusive finance. The information asymmetry between farmers and banks leads to adverse selection and moral hazard in credit rationing [2], which further restricts the development of rural credit market. For a long time, low-income people, especially poor households, have a higher degree of information asymmetry and stronger credit constraints, resulting in rural formal credit being squeezed by informal credit, and

farmers' credit demand in poor areas is more likely to be realized through informal credit channels [3]. In fact, the informal credit market is often accompanied by unfavorable factors such as high interest rates and underground banks, which will affect the stability of China 's rural financial market. At the same time, it will also be accompanied by a series of problems such as the " capture " of rural finance by the elite [4], which can not play the effectiveness of inclusive finance and hinder the stable development of rural economy and society [5]. In order to effectively solve the problem of high credit constraints of low-income groups and its derivative problems, microfinance has been rapidly promoted and developed in China since the 1990s [6]. Since the poverty alleviation, the poverty relief microfinance has become an effective means for China's financial precision to help poor households out of poverty. Since entering the transition period (2021–2025), the microcredit for the poverty-alleviated population has continued the support model of poverty relief microfinance [7], becoming an important financial support policy.

As an important part of rural finance, microcredit plays an important role in promoting agricultural and rural development and alleviating rural poverty. Since the implementation of the microcredit policy, the academic research on the evaluation of its implementation effect has been emerging. The research conclusions on whether access to microfinance can increase household production and operation income, alleviate poverty and improve livelihoods are controversial. Earlier studies have found that the implementation of microfinance has not effectively alleviated household poverty [8], especially for low-income groups with special difficulties, as well as rural households with high population burden ratio, the effect of microfinance on income growth is limited [9]. The main reason is that the special hardship group regards bank loans as a serious debt burden. If there are sudden natural disasters or external events, it may lead to the inability of low-income groups to repay loans [10–11]. However, in recent years, the academic community has generally recognized the role of microcredit in alleviating poverty and promoting income growth. In developing economies with imperfect markets and widespread resource constraints, microfinance policies have lowered the entry threshold for rural lenders to obtain formal financial loans and eased financial exclusion [12–13], to help farmers solve the problem of lack of funds, not only directly enriched the initial funds for farmers to engage in production and operation activities [14], but also indirectly reduced the risk of household consumption, ensured that loan households had sufficient funds in the process of development, production and operation, effectively revitalized various factors of production, especially alleviated the inequality of opportunities for low-income groups and improved women 's social participation [15], stimulate the endogenous development momentum of farmers, improve productivity and technical efficiency, and promote the increase of farmers 'production and operation income [16–17], which has the effect of reducing poverty and increasing income [18–19]. In addition, there is heterogeneity in the effect of microcredit on the income increase of farmers at different income levels. Wang Wencheng and Zhou Jinyu (2012) found that the effect of microcredit on the income increase of high-income farmers and low-income farmers was not significant, but the effect on the income increase of middle-income farmers was significant [20]. Wu Yan (2022) found that the impact of microcredit on high-income poverty-alleviated households at 0.75 quantile was significantly higher than that of poverty-alleviated households at 0.5 quantile and 0.25 quantile, showing the characteristics of income from high to low and the impact from strong to weak [21].

The above research provides a reference for this article, but there are also shortcomings, mainly as follows: At present, most of the research focuses on the income effect of poverty relief microfinance during the period of poverty alleviation, while there are few studies on the income effect of microcredit for the poverty-alleviated population in the period of consolidating the achievements of poverty alleviation and the effective connection of rural revitalization. In the new era, the monitoring object has become the focus of dynamic monitoring and assistance work to prevent poverty from returning to poverty. The monitoring object is composed of three types: unstable households, Marginal poverty-prone households and households with sudden and serious difficulties. Most of the research only focuses on the income effect of the microcredit policy of the poverty-alleviated population on the households that have been lifted out of poverty, while there are few studies on the income effect of the monitoring object. On the basis of the existing research, this paper focuses on the sample of the monitoring object, and re-evaluates the effect of the microcredit policy for the poverty- alleviated population. In terms of research methods, In terms of research methods, in the past research on evaluating the

effect of microfinance policy, the instrumental variable method, propensity score method (PSM) and other methods were mainly used to deal with the problems of endogeneity and sample selection bias. In order to better deal with the sample selection bias and solve the endogeneity problem and ensure that the research results are more reliable, this study selects the endogenous switching regression model. On the basis of existing research, this paper subdivides different types of poverty-alleviated population (as shown in Table 1), and based on the key work in the new era, focuses on the two types of households that have lifted out of poverty and monitoring objects, and based on the micro-survey data of six poverty- alleviated counties (cities) in an underdeveloped region in northwest China, this paper uses the endogenous transformation regression model to explore the income effect and heterogeneity of the microcredit for the poverty-alleviated population in China, and considers the impact of the microcredit for the poverty-alleviated population on the production and operation income of farmers under different initial endowment conditions. At the same time, it explores the mechanism path of the micro-credit policy for the poverty-stricken population to increase income. It not only enriches the academic research results related to inclusive finance for low-income people, but also provides a policy basis for further guiding China 's rural financial precision assistance and improving the efficiency of financial funds.

## 3. Theoretical analysis and research hypothesis

In the traditional rural financial system of China, low-income farmers are subject to credit constraints such as insufficient effective guarantee and mortgage, weak repayment ability, and high credit risk, and cannot obtain formal credit loans. During the period of poverty alleviation, the Chinese government has tailored the poverty alleviation microfinance policies, products and services for the development of poverty alleviation industries for poor households with filing cards, which are "less than 50,000 yuan, within three years, free of mortgage and guarantee, benchmark interest rate lending, poverty alleviation fund discount, and county construction risk compensation", and effectively solved the problems of poor households "unable to get loans", "poor use", "not yet available", and "difficult to sustain". In 2021, the China Banking and Insurance Regulatory Commission, the Ministry of Finance, the People 's Bank of China, and the National Rural Revitalization Bureau jointly issued the " Notice on In-depth and Solid Microcredit Work for Poverty Alleviation Population in the Transition Period, " which clearly stipulates that lenders must adhere to household borrowing, household use, and household repayment, and accurately use it for lenders to develop production and conduct operations. Therefore, when lenders use microcredit for the poverty-alleviated population, they must be used to develop production and carry out operations. When farmers obtain microcredit for the poverty-alleviated population, with the increase of capital factor input, the net income brought by capital factor is increasing. At the same time, compared with the traditional microcredit, the microcredit of the poverty-alleviated population is completely subsidized by the government, coupled with a more relaxed risk compensation mechanism, which eliminates the debt pressure of the borrower to a certain extent, and has a positive effect on encouraging the borrower to develop production and conduct business (Fig 1).

**Table 1. Differentiation criteria of different types of rural households in China.**

| Households type | Distinguish standard |
|---|---|
| General household | It refers to families with stable income and no risk of returning to poverty. |
| Households that have lifted out of poverty | It refers to those families that were once below the poverty income line set by the Chinese state, but have lifted out of poverty through poverty alleviation. |
| Monitoring object | The households that are unstable in poverty alleviation: It refers to the per capita net income within the scope of poverty prevention monitoring, and affected by various reasons there is a risk of returning to poverty, is included in the monitoring of poverty- alleviated households. |
| | Marginal poverty-prone households: It refers to the per capita net income within the scope of poverty prevention monitoring, and affected by various reasons there is a risk of poverty, is included in the monitoring of the general households. |
| | The households with sudden serious difficulties households: It refers to the per capita net income beyond the scope of monitoring, but affected by various factors such as emergencies led to a large rigid expenditure or a significant reduction in income, resulting in serious difficulties in basic life and the risk of returning to poverty, which were included in the monitoring and assistance of farmers. Such groups can be poverty alleviation households or general farmers, including households that do not enjoy the policy of poverty alleviation or monitoring objects that have eliminated risks. |

Farmers 'access to microcredit for poverty-alleviated population can not only alleviate the pressure of lack of cyclical funds caused by the development of production and operation, but also have more funds for the purchase of production materials, including seeds, fertilizers, agricultural machinery, etc., and can also lease land, purchase social services, etc., which helps to revitalize the production and operation activities of the lenders. In addition, in rural areas with relatively weak mobility and deeply influenced by traditional culture, social interaction between people has always been an important factor affecting farmers ' economic activities and their performance [22]. Farmers 'access to microcredit for poverty-alleviated population will enable them to have more funds for expanding social networks and strengthening social capital, so as to obtain more market information and technical information, reduce information asymmetry in the process of production and operation, and help improve production and operation income. In addition, compared with the traditional microcredit, the microcredit for poverty-alleviated population is completely subsidized by the government, coupled with a more relaxed risk compensation mechanism, which eliminates the debt pressure of the borrower to a certain extent, and has a positive effect on encouraging the borrower to develop production and conduct business.

In order to further clarify the theoretical research mechanism of the impact of micro-credit policy on farmers ' production and operation income, this study refers to the research of Wu Benjian et al. (2019), and constructs a theoretical model of the impact of micro-credit policy on farmers ' production and operation income [23]. It is assumed that the farmer 's production function is Cobb-Douglas form:

$$Y = AK^{\alpha}L^{\beta}$$

Among them, Y is the income of production and operation, K is the material capital, L is the labor force, A is the total factor productivity. Considering the limited land carrying capacity of farmers and the limited level of technical management, it is

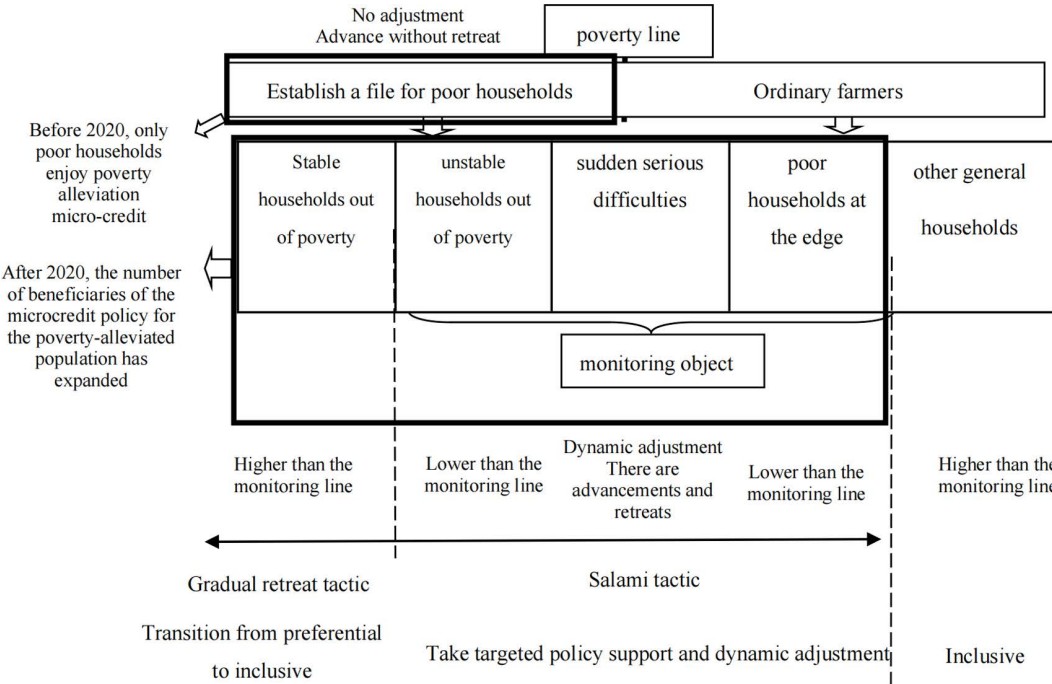

**Fig 1. The schematic diagram of China 's support policy system.**

assumed that the production function of farmers is diminishing returns to scale, that is, $\alpha + \beta < 1$, and $\alpha$, $\beta \in (0,1)$. and the budget constraint of farmers is:

$$K \leq K_0 + S, S = \begin{cases} s(\text{loaned}) \\ 0(\text{unloaned}) \end{cases}$$

Among them, $K_0$ is the initial capital, S is the credit funds. Income is calculated for farmers who obtained credit (S=s) and farmers who have not obtained credit (S=0) respectively:

Farmers who have not obtained credit: $Y_0 = AK_0^\alpha L^\beta$

Farmers who obtained credit: $Y_1 = A(K_0 + S)^\alpha L^\beta$

$$\Delta Y = Y_1 - Y_0 = AL^\beta \left[(K_0 + S)^\alpha - K_0^\alpha\right] > 0$$

Since $\alpha > 0$ and $s > 0$, credit expands the material capital input ($K_0 \to K_0 + s$) and significantly increases income.

Further expand the model and incorporate social capital into the production function:

$$Y = AK^\alpha L^\beta W^\theta$$

Among them, W is social capital (such as cooperative network, information acquisition ability), $\alpha$, $\beta$, $\theta \in (0,1)$, $\alpha + \beta + \theta < 1$.

The microcredit for poverty-alleviated population will directly affect the material capital investment and social capital investment, therefore:

$$K = K_0 + \gamma_1 C, W = W_0 + \gamma_2 C$$

Among them, $K_0, W_0$ are the initial capital, $\gamma_1, \gamma_2$ are the efficiency coefficients of credit into two types of capital, $\gamma_1, \gamma_2 \in (0,1)$.

Substituting K and S into the production function, we get:

$$Y = AL^\beta (K_0 + \gamma_1 C)^\alpha (W_0 + \gamma_2 C)^\theta$$

The partial derivative of credit to income can be obtained:

$$\frac{\partial Y}{\partial C} = AL^\beta \left[\alpha\gamma_1(K_0 + \gamma_1 C)^{\alpha-1}(W_0 + \gamma_2 C)^\theta + \beta\gamma_2(K_0 + \gamma_1 C)^\alpha(W_0 + \gamma_2 C)^{\theta-1}\right]$$

$$\frac{\partial Y}{\partial C} = AL^\beta \left[\alpha\gamma_1 K^{\alpha-1}W^\theta + \theta\gamma_2 K^\alpha W^{\theta-1}\right]$$

Because $\alpha, \beta, \gamma_1, \gamma_2 > 0$, $\frac{\partial Y}{\partial C} > 0$, that is, credit significantly increases income by increasing material and social capital investment. That is, if $C_{\text{loaned}} > C_{\text{unloaned}} = 0$, then $Y_{\text{loaned}} > Y_{\text{unloaned}}$.

Further, the total effect $\frac{\partial Y}{\partial C}$ is decomposed into two paths: material capital investment and social capital investment:

In terms of material capital investment: $Effect_K = A\alpha\gamma_1 L^\beta K^{\alpha-1}W^\theta > 0$

In terms of social capital investment: $Effect_W = A\theta\gamma_2 L^\beta K^\alpha W^{\theta-1} > 0$

The total effect satisfies: $\frac{\partial Y}{\partial C} = Effect_K + Effect_W > 0$

This shows that microfinance can promote farmers to increase material capital investment and social capital investment, and further increase production and operating income.

Based on the above theoretical analysis and derivation, the following two research hypotheses are proposed:

**Hypothesis 1:** Compared with farmers who do not obtain microcredit for the poverty-alleviated population, obtaining microcredit for the poverty-alleviated population can promote the increase of their production and operation income. That is, microcredit for the poverty-alleviated population has an income-increasing effect.

**Hypothesis 2:** The microcredit policy for poverty-alleviated population promotes the increase of production and operation income by promoting farmers to increase material capital investment and social capital investment.

The precise implementation of the microcredit policy for the poverty-alleviated population should take into account the initial resource endowment level of farmers. Under the goal of common prosperity, the focus of rural revitalization has shifted from eliminating absolute poverty to alleviating relative poverty. At this stage, how to improve the endogenous development momentum of the poverty-alleviated population and how to achieve the goals of ' raising the low ' and ' expanding the middle ' has become an important task to alleviate relative poverty. In fact, microcredit for poverty-alleviated population mainly helps low-income groups with poor resource endowments to solve financing difficulties [7]. No matter during the period of poverty alleviation or the period of comprehensively promoting rural revitalization, microcredit has played a ' hematopoietic ' role in promoting the production and operation of poverty-alleviated households and monitoring objects. However, in the new era, some poverty-alleviated households and monitoring objects have leaped from low-income groups to middle-income groups through long-term assistance and self-development, and their own resource endowments have been at a high level. In theory, the marginal effect of credit capital input on increasing farmers' income is decreasing [24]. Therefore, for poverty-alleviated households and monitoring objects with poor resource endowments, the higher the income of credit capital input to promote farmers ' production and operation, and for poverty-alleviated households and monitoring objects with higher resource endowments, the effect of credit capital investment on increasing farmers' income is relatively small. That is to say, the microcredit policy for the poverty-alleviated population has a greater income-increasing effect on the lower-income poverty-alleviated households and monitoring objects engaged in production and operation. Therefore, the following hypothesis is proposed:

**Hypothesis 3:** There are differences in the income effect of microcredit for poverty-alleviated population on farmers with different initial endowments, and the income effect on farmers with low initial endowments is higher than that on farmers with high initial endowments. That is, the microcredit for the poverty-alleviated population has a ' raising the low ' effect.

## 4. Data source, descriptive analysis and model construction

### 4.1 Data sources

The data in this paper are derived from the sample survey data of farmers in six poverty-alleviated counties (cities) in an underdeveloped area of northwest China, in January 2023. This region is one of the originally concentrated and contiguous areas of deep poverty in China, and it is also the ethnic border area in northwest China. It has a high degree of particularity in terms of geographical location, resource endowment, ecological environment, cultural background, development stage and development ability. The scale of poverty-alleviated population in this area is large and the proportion is high. This region is not only a daunting task for China to consolidate its poverty alleviation achievements, but also one of the main battlefields for consolidating poverty alleviation achievements and effectively connecting with rural revitalization in the northwest region.By the end of 2020, 984,100 (37.43% of the poverty rate at the end of 2013) rural poor people in  this area were all lifted out of poverty. As of January 2022, the area has issued a total of 363,800 households with a total of 12.757 billion yuan of microcredit for the poverty-alleviated population. Financial precision assistance has played a pivotal role in this area. Under the policy of " four non-picking " in the new era, it is typical and representative to select the sample of farmers in the border areas of northwest China and the original deep poverty areas .Based on the comprehensive factors such as geographical location, economic development level, number of monitoring objects, and village attributes, this survey randomly selected 10 villages in 5 townships in each of the six poverty-alleviated counties (cities), including 8 poverty-stricken villages (including large and medium-sized resettlement sites) and 2 general villages with relatively heavy rural revitalization tasks. About 22 households were selected on average in each village, and a total of 1350 questionnaires were collected.

According to the roster of farmers provided by the sampling village, considering the family income status, health status, disability and so on, a stratified multi-stage survey was carried out in the sampling village. This paper mainly focuses on the survey of the households that have lifted out of poverty, monitoring objects and general households with low income, the proportion of these three types of farmers 'sampling samples is roughly 25%, 30% and 45%. collecting and mastering the income status, assistance, microcredit, industrial development and rural governance and construction of the survey objects. According to the policy, the borrowing objects of microcredit for the poverty-alleviated population must be poverty-alleviated households and monitoring objects, and must be used for the development of production and operation. Therefore, this paper excludes the general households and the farmers without production and operation income. The effective sample is 738, of which 329 households have been lifted out of poverty. The monitoring objects are 409 households (including 54 households with sudden serious difficulties, 183 households with unstable poverty alleviation, and 172 households with marginal poverty). The distribution of survey samples is shown in Table 2. The households that have lifted out of poverty refer to the original file cardholders with higher income levels and no risk of returning to poverty at the current stage. The monitoring objects refer to the low-income households that have experienced a sharp increase in expenditure or a sharp decrease in income due to illness, sudden accidents and other reasons since 2019, and at the time there was a risk of returning to poverty.

## 4.2 Variable definition and statistical characteristics

Explained variables. This paper selects the per capita production and operation income of the family as the explained variable. In the survey, the sample farmers were asked about the household operating income from planting, breeding, business, etc.in the whole year of 2022. According to the actual living population of the family, the per capita operating income of the household was calculated and the natural logarithm was taken as the explanatory variable of this article.

Core explanatory variables. In essence, this paper mainly discusses the income effect of microcredit for the poverty-alleviated population, so the ' whether to obtain microcredit for the poverty-alleviated population ' in the questionnaire is taken as the core explanatory variable, and this paper only studies the poverty-alleviated households and monitoring objects who meet the qualification of microcredit for the poverty-alleviated population and have developed production and operation.

Control variables. This paper controls the variables that may affect the income of farmers' production and operation, including the variables of household head characteristics (age, gender, education level), family characteristics variables (proportion of the elderly, proportion of young adults, proportion of disability, proportion of subsistence allowances), production and operation characteristics variables (land transfer in, land transfer out, skill training, distance from village to county).

Instrumental variables. Based on the selection of instrumental variables by Huo Yu et al. [25] and He Jing et al. [26], considering that the acquisition of microcredit for the poverty-alleviated population will be affected by the ' peer effect ', that is, it may be related to the behavior imitation among farmers of similar ages in the same village. So this article is grouped according to the village and age (40 years old and below, 41–50 years old, 51–60 years old, over 60 years old),

Table 2. Survey sample distribution.

| Household type | | Ht City | Het county | My county | CI county | Yt county | Mf county | total |
|---|---|---|---|---|---|---|---|---|
| The households that have lifted out of poverty. | | 63 | 42 | 60 | 62 | 51 | 51 | 329 |
| Monitoring object | marginal households that are prone to poverty | 32 | 19 | 24 | 24 | 36 | 37 | 172 |
| | The households that are unstable in poverty alleviation | 23 | 42 | 36 | 30 | 32 | 20 | 183 |
| | The households with sudden serious difficulties | 4 | 6 | 12 | 25 | 4 | 3 | 54 |
| total | | 122 | 109 | 132 | 141 | 123 | 111 | 738 |

and calculates the average level of farmers of similar age groups in the same village to obtain microcredit for the poverty-alleviated population as an instrumental variable, and in the extreme case where the average level is 0 or 1, this paper replaces it with the whole sample mean. Theoretically,

grouping by village can capture the differences in credit environment in different regions, while the age group of house-holders reflects the differences in credit demand and ability to use in different age groups. This grouping method can effectively separate the factors related to credit decision-making but not directly related to the final result (such as income level), so as to meet the exogeneity of instrumental variables. At the same time, as an instrumental variable, the average value of loans obtained can reflect the relative credit acquisition of individuals in the group, which is related to the individual credit decision-making, but it avoids the interference of direct causality and helps to solve the endogenous problem.

Table 3 is the definition and assignment of all variables in this paper. It can be seen from Table 3 that the production and operation income of farmers who obtain microcredit for poverty-alleviated population is higher, which indicates that obtaining microcredit for poverty-alleviated population may increase the production and operation income level of farmers. In terms of other variables, the farmers who have obtained microcredit for poverty-alleviated people show higher levels of household head characteristics such as gender and education level, the characteristics of families such as the actual population and the proportion of young and middle-aged people, and the characteristics of production and operation such as social networks and skills training. In addition, farmers who do not obtain microcredit for the poverty-alleviated population show a high level of household head and family characteristics such as age, proportion of the elderly, proportion of disability, proportion of subsistence allowances, etc. Although Table 3 intuitively reflects that there are differences in the mean values of some variables under whether or not to obtain microcredit for poverty-alleviated population, it does not indicate that these differences are caused by whether or not to obtain microcredit for p poverty-alleviated population. In order to accurately demonstrate the income effect of microcredit for the poverty-alleviated population, it is also necessary to fully consider the selective bias caused by the ' self-selection ' of the sample. Therefore, the following uses a more scientific endogenous conversion regression model for empirical analysis.

## 4.3 Model construction

### 4.3.1 Endogenous transformation regression model (ESR).

Whether farmers can obtain microcredit for poverty-alleviated population and the income of their production and operation is also constrained by their own resource endowments, such as the number of family labor, land resources, industrial base, social capital and so on. At the same time, there are also unobservable factors that may affect farmers ' access to microcredit for poverty-alleviated population and the income of production and operation, such as cognitive ability, learning ability, support strength, etc. These unobservable factors will lead to sample selection bias and endogenous problems. There may be a problem of ' simultaneous decision-making ' between the loans decision-making of microcredit for the poverty-alleviated population and the income effect it brings. If models are established separately for estimation, it may also lead to inaccurate research conclusions due to selective bias and endogenous problems caused by unobservable factors [27]. Although the propensity score matching method can solve the problem of sample bias and variable endogeneity, it cannot solve the sample selection bias caused by unobservable factors [28]. The endogenous transformation regression model can simultaneously deal with the impact of observable and unobservable factors on farmers' decision-making to obtain microcredit for the poverty-alleviated population. By building a counterfactual framework, the conditional expectations of the production and operation income of farmers who obtain and do not obtain microcredit for the poverty-alleviated population are analyzed, and then the average treatment effect of farmers' access to microcredit for the poverty-alleviated population on production and operation income is further analyzed, revealing the real impact of access to microcredit for the poverty-alleviated population on farmers ' production and operation income.

The basic idea of the ESR model is to evaluate it in two stages. The first stage is the behavioral decision equation, that is, to estimate the influencing factors of whether farmers have access to microcredit for the poverty-alleviated population;

Table 3. Variable definition and descriptive statistics.

| | variable | Variable definition and assignment | Average sample of poverty-stricken households | | Mean value of monitoring object samples | | Overall sample mean | |
|---|---|---|---|---|---|---|---|---|
| | | | Unloaned households(N=107) | Loaned households(N=222) | Unloaned households(N=165) | Loaned households(N=244) | Unloaned households(N=272) | Loaned households(N=466) |
| **Explained variable** | The natural logarithm of production and operation income | In order to weaken the influence of extreme values, the natural logarithm of per capita production and operation income is taken. | 7.851 | 8.121 | 7.743 | 7.909 | 7.786 | 8.01 |
| **Core explanatory variables** | Microcredit for poverty-alleviated population | The micro-credit for the poverty-alleviated population has been obtained=1; unacquired population microcredit=0 | 0 | 1 | 0 | 1 | 0 | 1 |
| **Householder characteristic variable** | Sexuality | Male=1; female=0 | 0.86 | 0.901 | 0.77 | 0.893 | 0.805 | 0.897 |
| | Age | The actual age of the head of household | 57.523 | 50.788 | 53.897 | 47.586 | 55.324 | 49.112 |
| | Standard of culture | Illiterate and semi-illiterate=1; primary school=2; junior high school=3; high school/ technical secondary school=4; university and above = 5 | 2.234 | 2.419 | 2.37 | 2.471 | 2.316 | 2.446 |
| **Family characteristic variables** | Proportion of elderly | Number of people aged 60 and above/ actual population | 0.264 | 0.082 | 0.222 | 0.059 | 0.239 | 0.07 |
| | Proportion of young adults | Number of persons aged 16–59 years/ actual population | 0.499 | 0.583 | 0.478 | 0.57 | 0.486 | 0.576 |
| | Proportion of disability | Number of disabled people/ actual population | 0.028 | 0.046 | 0.089 | 0.022 | 0.065 | 0.033 |
| | Proportion of subsistence allowances | Number of subsistence allowances/ actual population | 0.363 | 0.31 | 0.575 | 0.443 | 0.491 | 0.38 |
| **Production and operation characteristic variables** | Agricultural land transfer-out | Land transfer=1, no land transfer=0 | 0.121 | 0.081 | 0.103 | 0.086 | 0.11 | 0.084 |
| | Agricultural land transfer-into | Land transfer out = 1, no land transfer out = 0 | 0.121 | 0.054 | 0.036 | 0.053 | 0.07 | 0.054 |
| | Competence training | Have participated in skill training=1, have not participated in skill training=0 | 0.159 | 0.266 | 0.133 | 0.27 | 0.143 | 0.268 |
| | Distance | Distance from the village committee to the county (km) | 37.15 | 37.599 | 31.497 | 38.508 | 33.721 | 38.075 |
| **instrumental variable** | Average microcredit access rate | The average level of farmers of similar age groups in the same village to obtain microcredit for the poverty-alleviated population, and the extreme situation with an average level of 0 or 1 is replaced by the full sample average. | 0.535 | 0.65 | 0.538 | 0.661 | 0.537 | 0.656 |

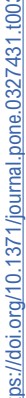

the second stage is the result equation, that is, the production and operation income of the ' loaned ' and ' non-loaned ' groups of farmers is estimated to study the differences in the production and operation income of the loan households under different circumstances.

The behavioral decision-making equation of whether to obtain microcredit for the poverty-alleviated population is constructed as follows:

$$S_i^* = \pi_i Z_i + \theta_i I_i + \mu_i \tag{1}$$

In the formula, $S_i^*$ represents the latent variable of whether the farmer i obtains the microcredit of the poverty-alleviated population. When $S_i = 1$, it means that the farmer obtains the microcredit of the poverty-alleviated population. When $S_i = 0$, it means that the farmer doesn't obtain the microcredit of the poverty-alleviated population. $Z_i$ represents the exogenous explanatory variable that affects whether the farmer obtains the microcredit of the poverty-alleviated population. $I_i$ represents the identification variable, $\pi_i$、$\theta_i$ represents the coefficient to be estimated, and $\mu_i$ represents the random disturbance term.

Therefore, the decision equation of farmers 'production and operation income can be expressed as:

$$Y_{1i} = X_{1i}\beta_1 + \sigma_{1\mu}\lambda_{1i} + \varepsilon_{1i}\ (S_i = 1) \tag{2}$$

$$Y_{0i} = X_{0i}\beta_0 + \sigma_{0\mu}\lambda_{oi} + \varepsilon_{0i}\ (S_i = 0) \tag{3}$$

Eq. (2) represents the estimation equation of the production and operation income of the farmers who have obtained the microcredit for poverty-alleviated population, and Eq. (3) represents the estimation equation of the production and operation income of the farmers who have not obtained the microcredit for poverty-alleviated population. $Y_{1i}$ And $Y_{0i}$ respectively represent the production and operation income of the farmers who have obtained and have not obtained the microcredit for poverty-alleviated population, $X_{1i}$ and $X_{0i}$ respectively represent the factor variables that affect the production and operation income of the farmers in the two situations, $\varepsilon_{1i}$ and $\varepsilon_{0i}$ represent the random disturbance term, $\beta_1$、$\beta_0$ represent the coefficient to be estimated. In order to further solve the problem of self-selection bias caused by unobservable variables, the inverse Mills ratio $\lambda_{1i}$、$\lambda_{0i}$ and its covariance $\sigma_{1u} = cov\,(\mu_i, \varepsilon_{1i})$、$\sigma_{0\mu} = cov\,(\mu_i, \varepsilon_{0i})$ are introduced, and the simultaneous estimation of Eq. (1) to Eq. (2) is carried out by using the complete information maximum likelihood method.

By comparing the production and operating income expectations of whether or not to obtain microcredit for the poverty-alleviated population under the factual and counterfactual scenarios, the average processing effect of whether or not to obtain microcredit for the poverty-alleviated population on the production and operating income of farmers is obtained. The estimation process is as follows:

The expected value of the production and operation income of the farmers who have obtained microcredit for the poverty-alleviated population:

$$E[Y_{1i}|S_i = 1] = X_{1i}\beta_1 + \sigma_{1\mu}\lambda_{1i} \tag{4}$$

The expected value of production and operation income of farmers who have not obtained the microcredit for poverty-alleviated population:

$$E[Y_{0i}|S_i = 0] = X_{0i}\beta_0 + \sigma_{0\mu}\lambda_{0i} \tag{5}$$

In the counterfactual situation, the expected value of the production and operation income of the farmers who have obtained the microcredit for poverty-alleviated population in the non-acquisition situation:

$$E[Y_{0i}|S_i = 1] = X_{1i}\beta_0 + \sigma_{0\mu}\lambda_{1i} \tag{6}$$

The expected value of production and operation income of farmers who have obtained microcredit for poverty-alleviated population in the context of access:

$$E[Y_{1i}|S_i = 0] = X_{0i}\beta_1 + \sigma_{1\mu}\lambda_{0i} \tag{7}$$

Through Eq. (4) and Eq. (6), the average treatment effect of farmers 'actual access to the production and operation income of microcredit for the poverty-alleviated population is:

$$ATT_i = E[Y_{1i}|S_i = 1] - E[Y_{0i}|S_i = 1] = X_{1i}(\beta_1 - \beta_0) + (\sigma_{1\mu} - \sigma_{0\mu})\lambda_{1i} \tag{8}$$

Similarly, through Eq. (5) and Eq. (7), the average treatment effect of the production and operation income of the farmers who actually have not obtained the microcredit for the poverty-alleviated population is:

$$ATU_i = E[Y_{1i}|S_i = 0] - E[Y_{0i}|S_i = 0] = X_{0i}(\beta_1 - \beta_0) + (\sigma_{1\mu} - \sigma_{0\mu})\lambda_{0i} \tag{9}$$

**4.3.2 Quantile regression model.** Based on the analysis of the effect of the production and operation income brought by the microcredit for the poverty-alleviated population. this paper draws on the research ideas of Huo Yu et al. [26], and uses the quantile regression model to further estimate the impact of obtaining microcredit for poverty-alleviated populations on the production and operating income of farmers with different initial endowments. Taking the income of farmers ' production and operation (natural logarithm) as the explained variable, whether or not farmers obtain microcredit for the poverty-alleviated population and the characteristics of individual, family, production and operation as the explanatory variables, the following quantile regression model is established:

$$Y_\tau(Y|S) = \chi_\tau + \varphi_\tau S_i + \omega_\tau X_i + \upsilon_\tau \tag{10}$$

In Eq.(10), $Y_\tau (Y|S)$ indicates the production and operation income of farmers in the $\tau$ quantile, S indicates the access to microcredit for the poverty-alleviated population, i indicates the i-th farmer, $X_i$ indicates the factors affecting the production and operation income of farmers in addition to the microcredit for the poverty-alleviated population, $\chi_\tau$ indicates the constant term, $\varphi_\tau$、 $\omega_\tau$ indicates the coefficient to be estimated, and $\upsilon_\tau$ indicates the random disturbance term.

## 5. Empirical test and result analysis

### 5.1. The estimated results of the impact of obtaining microcredit for the poverty-alleviated population on the income of farmers' production and operation

Table 4 is the estimation results of the ESR model for the impact of obtaining microcredit for the poverty-alleviated population on farmers ' production and operation income. From Table 4, we can see that the correlation coefficient of the error term of the simultaneous estimation of the microcredit of the poverty-alleviated population and the income of the farmers 'production and operation income, ρ1 is significant at the statistical level of 1%, indicating that there is indeed a self-selection bias in the sample. If not corrected, it will lead to biased estimation results. ρ1 is negative, indicating that the production and operation income of farmers who obtain microcredit for poverty-alleviated population is higher than that of random individuals in the sample farmers. This conclusion shows that obtaining microcredit for poverty-alleviated population can improve the production and operation income of farmers. In addition, the two-stage equation independence LR test in the model rejects the null assumption that the decision-making equation and the result equation are independent at the 1% level, and the Wald test of goodness of fit is significant at the level of 1%, indicating that the endogenous transformation regression model is

**Table 4. ESR model estimation results of the impact of microcredit on farmers' production and operation income.**

| Variable | Loan decision equation | Result equation | |
|---|---|---|---|
| | | Loaned | Unloaned |
| Sexuality | 0.233* | 0.0678 | −0.184 |
| | (0.138) | (0.168) | (0.170) |
| Age | −0.000348 | 0.00198 | 0.00280 |
| | (0.00555) | (0.00652) | (0.00663) |
| Standard of culture | 0.00869 | −0.0509 | −0.126 |
| | (0.0932) | (0.106) | (0.121) |
| Proportion of elderly | −1.034*** | 1.769*** | 0.627* |
| | (0.277) | (0.353) | (0.335) |
| Proportion of young adults | 0.105 | 0.264 | 0.293 |
| | (0.175) | (0.201) | (0.235) |
| Proportion of disability | −0.486 | 0.972* | 0.0962 |
| | (0.364) | (0.496) | (0.351) |
| The proportion of subsistence allowances | 0.0510 | −0.752*** | −0.585*** |
| | (0.156) | (0.183) | (0.194) |
| Agricultural land transfer-out | −0.174 | 0.207 | 0.0276 |
| | (0.173) | (0.197) | (0.227) |
| Agricultural land transfer-into | −0.242 | −0.261 | −0.206 |
| | (0.205) | (0.230) | (0.249) |
| Competence training | 0.150 | −0.244* | −0.284 |
| | (0.119) | (0.125) | (0.189) |
| Distance | −0.000851 | 0.00475*** | 0.00973*** |
| | (0.00118) | (0.00136) | (0.00160) |
| Instrumental variable | 1.498*** | | |
| | (0.282) | | |
| Constant term | 8.642** | 8.642*** | 7.801*** |
| | (0.538) | (0.538) | (0.686) |
| $\rho_1$ | | −0.935*** | |
| | | (0.0296) | |
| $\rho_0$ | | | 0.0173 |
| | | | (0.309) |
| LR test of equation independence | | 24.52*** | |
| Logarithmic likelihood value | | −1440.3032 | |
| Goodness of fit Wald test | | 82.91*** | |
| Sample size | | 738 | |

Note: *, * *, * * * are expressed at a significant level of 10%, 5%, 1%, respectively, the same below.

reasonable. In the loan decision equation of Table 4, the instrumental variable is significantly positive at the statistical level of 1% for obtaining microcredit for the poverty-alleviated population. In order to further verify the effectiveness of the instrumental variable, referring to the research ideas of Li Jiahui and Lu Qian [27], in the case of introducing control variables, OLS regression is carried out with the situation of microcredit access to the poverty-alleviated population and instrumental variable as independent variables, and the production and operation income of farmers as the dependent variable. The results show that the instrumental variable has no significant impact on the production and operation income of farmers. In addition, this study also uses the instrumental variable method (IV-2SLS) for empirical testing. The results show that the first stage F value

is 17.87, which is greater than the empirical value of 10, indicating that the instrumental variable is not a weak instrumental variable. Therefore, it can be considered that the instrumental variables selected in this study are effective.

### 5.1.1. The estimation results of the influencing factors of farmers' access to microcredit for poverty-alleviated population.

Among the family characteristic variables, The estimated coefficient of gender in the loan decision equation is 0.233, and it is significant at the statistical level of 10%, indicating that families with male heads of households will significantly increase the probability of farmers obtaining microfinance for the poverty-alleviated population. Generally speaking, in China, the male head of the household is male, indicating that the family is the main decision-maker. Compared with female decision-makers, men tend to be more rational and forward-looking, and have a higher degree of mastery and acceptance of policies than women. Therefore, households headed by men will increase the probability of obtaining microcredit for people out of poverty. the estimated coefficient of the proportion of the elderly in the loan decision equation is −1. 034, and it is significant at the statistical level of 1%, indicating that the higher proportion of the elderly in the family will significantly reduce the probability of farmers obtaining microcredit for the poverty-alleviated population. The possible reason is that the higher the proportion of the elderly, the less the labor force in the family, and the limited labor force can't further expand the scale of production and operation, resulting in a relatively small demand for funds to expand the scale of production and operation. Therefore, the households that have lifted out of poverty and monitoring objects with heavy pension burden will reduce the probability of obtaining microcredit for the poverty-alleviated population.

### 5.1.2. The estimated results of the impact of obtaining microcredit for the poverty-alleviated population on the income of farmers' production and operation.

In the family characteristic variables, the proportion of the elderly and the proportion of the disabled have a significant positive impact on the production and operation income of the farmers who obtain the microcredit of the poverty-alleviated population. Families with a high proportion of the elderly or the disabled will generally have the adult labor force to take care of the elderly or the disabled. Although the burden of such families is heavy, the adult labor force still has the ability to develop production and operation. Therefore, with the help of microcredit for the poverty-alleviated population, it can stimulate such families to expand the scale of production and operation, and then increase the production and operation income. The proportion of subsistence allowances has a significant negative impact on the production and operation income of farmers who obtain and do not obtain microcredit for the poverty-alleviated population. Under the precise assistance policy, the higher the proportion of subsistence allowances, the less the family's labor force on behalf of the family, and the less the labor force required for the development of production and operation. Therefore, regardless of whether they are loaned or not, the proportion of subsistence allowances has a negative impact on their production and operation income. In terms of production and operation characteristics, skill training has a significant negative impact on the production and operation income of farmers who obtain microcredit for the poverty-alleviated population. Farmers who participate in skill training not only help to improve their own skill level, but also broaden their knowledge in the training process. And skill training in the region mainly focuses on employment-oriented e-commerce, beauty salons, brick and tile workers, planting and breeding, etc. Therefore, skill training has greatly promoted the transfer of local rural labor to higher-yielding non-agricultural sectors with higher returns. So skill training has a significant negative impact on the production and operation income of loaned households. Distance has a significant positive impact on the production and operation income of farmers who have obtained and have not obtained microcredit for poverty-alleviated population. The villages closer to the county have significant advantages in transportation, service industry, factory and other aspects. The more employment opportunities the farmers have, the less investment they will make in the development of production and operation. vice versa. Therefore, the farther away from the county, the higher the income of production and operation.

### 5.2. The analysis on the treatment effect of microcredit for poverty-alleviated population on the income of farmers' production and operation

The estimation results of the treatment effect of farmers' access to microcredit for poverty-alleviated population on their production and operation income are as shown in Table 5. The average treatment effect of farmers' access to microcredit

for poverty -alleviated population on their production and operation income is significantly positive at the statistical level of 1%. The estimation results of ATT show that under the counterfactual assumption, if farmers who obtain microcredit for poverty-alleviated population do not receive loans (counterfactual), their production and operation income will decrease from 8.017 to 7.752, a decrease of 3.31%; the estimation results of ATU show that under the counterfactual hypothesis, if the farmers who have not obtained the microcredit of the poverty-alleviated population get the loan (counterfactual), their production and operation income will rise from 7.786 to 9.893, an increase of 27.06%. According to this, it can be seen that the acquisition of micro-credit for the poverty-alleviated population has significantly increased the income of farmers ' production and operation. If the farmers who do not obtain micro-credit for the poverty-alleviated population are loaned, the income of their production and operation income is more obvious, which can increase the income of farmers' production and operation by 3.31%~27.06%.

In order to more clearly reflect the income effect of microcredit for poverty-alleviated population, the probability density distribution maps of production and operating income of farmers who have obtained and have not obtained microcredit for poverty-alleviated population are described respectively(Fig 2A and 2B). The ATT of the left figure shows that if farmers who have obtained microcredit for poverty-alleviated population have not been loaned, the probability density distribution curve of their production and operating income is obviously shifted to the left. Lenders are in the 'counterfactual 'situation that they do not obtain microcredit for the poverty- alleviated population, the production and operating income of farmers is significantly reduced(Fig 2A). The ATU of the right figure shows that the probability density distribution curve of the production and operation income of the uncredited households in the ' counterfactual ' situation of obtaining microcredit for the poverty-alleviated population has shifted significantly to the right, indicating that if the uncredited households obtain

**Table 5. The average treatment effect of microcredit for poverty-alleviated population on farmers' production and operating income.**

| Group | Decision stage | | Average treatment effect | |
|---|---|---|---|---|
| | Loaned | Unloaned | ATT | ATU |
| Loaned Group(N = 466) | 8.017 | 7.752 | 0.265*** | — |
| | (0.0184) | (0.0236) | (0.0299) | |
| Unloaned group(N = 272) | 9.893 | 7.786 | — | 2.107*** |
| | (0.03) | (0.0338) | | (0.0452) |

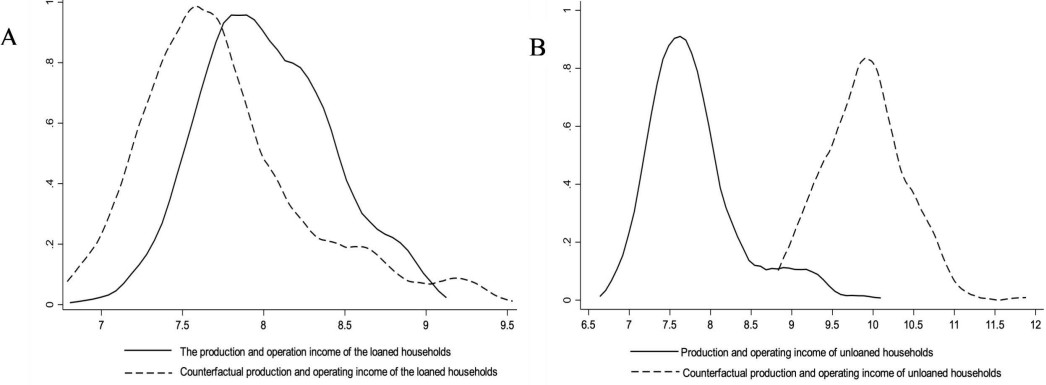

**Fig 2. Probability density of farmers' production and operation income in two cases.** (A) Income Change Chart of Rural Households Eligible for Poverty Alleviation Microcredit but Not Granted Loans (Counterfactual).(B) Income Change Chart of Rural Households Not Eligible for Poverty Alleviation Microcredit but Hypothetically Granted Loans (Counterfactual).

microcredit for the poverty-alleviated population, their production and operation income will increase significantly(Fig 2B), therefore, the research hypothesis 1 is verified. In fact, the microcredit for poverty alleviated population is a preferential policy for low-income farmers. Through measures such as financial discounts and interest rate concessions, the financing costs of poverty-alleviated households and monitoring object are reduced. It can not only help poverty-alleviated households solve financial bottlenecks and support their development of production or operation, but also help to enhance the self-development ability, enhance their economic autonomy, and promote their integration into the development of the market economy, thereby increasing production and operating income. However, the farmers who have not actively applied for microcredit for poverty-alleviated population, their development is relatively lack of financial support, and it is difficult to carry out or expand production and operation activities, resulting in limited income growth.

### 5.3. Robustness test

In order to verify the reliability of the above research conclusions, this paper uses three methods to test the robustness of the effect of income increase on the microcredit for poverty-alleviated population by replacing the empirical method, tail-down processing, and replacing the explanatory variables, and obtains Table 6.

Robustness test method 1: Replace the empirical method. In this paper, the propensity score matching (PSM) method is used to estimate the effect of the microcredit for poverty-alleviated population on the income of farmers' production and operation. The farmers who obtained microcredit for the poverty-alleviated population were taken as the control group, and the farmers who did not obtain microcredit for poverty-alleviated population were taken as the treatment group, the nearest neighbor matching (K = 4) is selected for matching. This part only reports the estimation results of the PSM model. The ATT value of the impact of microcredit for poverty-alleviated population on the income of farmers ' production and operation is 0.279, indicating that the microcredit for poverty-alleviated population has a significant effect on the income of farmers' production and operation, which is consistent with the estimation results of the ESR model. The difference is that the effect of microcredit for poverty-alleviated population on farmers' production and operation income obtained by PSM method is higher than that of ESR model. This is because the PSM model does not take into account the influence of unobservable factors, resulting in biased estimation results. The ESR model not only solves the problem of self-selection and endogeneity of microcredit for the poverty-alleviated population, but also fully considers the selective bias caused by observable and unobservable factors. The bias term obtained in the first stage is automatically added to the second stage to estimate the impact of microcredit for the poverty-alleviated population on the income of farmers' production and operation, and the estimation results are more scientific [29].

Robustness test method 2: Tailing treatment. In order to eliminate the influence of extreme values of data on the estimation results [30], referring to the ideas of Falck et al. [31] and Pan et al. [32], the method of tail treatment is adopted to eliminate the extreme values of 5% at the beginning and end of the data of farmers' production and operation income, and the model is estimated again. The ATT value of the impact of microcredit for poverty-alleviated population on farmers' production and operation income is 0.418, indicating that microcredit for poverty-alleviated population has a significant effect in promoting farmers' production and operation income, which is consistent with the above conclusions.

Robustness test method 3: replace the explained variable. In fact, after many farmers obtain microcredit for the poverty-alleviated population, they put the assets such as cattle and sheep purchased by the loan into companies or

**Table 6. Robustness test.**

| Robustness test method | Loaned | Unloaned | ATT | t-values |
|---|---|---|---|---|
| Propensity score matching | 8.007 | 7.728 | 0.279*** | 2.66 |
| Tailing processing | 8.028 | 7.609 | 0.418*** | 14.558 |
| Replace the explained variable | 8.102 | 7.76 | 0.342*** | 11.713 |

cooperatives for dividends, and a small number of farmers put the credit funds into banks or lending enterprises to obtain interest income. This part of the income is also generated by microcredit for the poverty-alleviated population, but it is property income in income statistics. In order to further demonstrate the income effect of microcredit for the poverty-alleviated population, the property income and production and operation income are added to form property and production and operation income, and the property and production and operation income are replaced with new explanatory variables. The ATT value of the impact of microcredit for poverty-alleviated population on farmers' production and operation income is 0.342, indicating that microcredit for poverty-alleviated population has a significant role in promoting farmers' production and operation income, which is consistent with the above conclusions.

## 5.4. Heterogeneity analysis

Considering the heterogeneity of the effect of income increase on the microcredit for the poverty-alleviated population between the households who have lifted out of poverty and the monitoring objects. Therefore, we further study the differences in the production and operating income of the two types of households that have lifted out of poverty and the monitoring objects. As shown in Table 7, from the perspective of households that have lifted out of poverty, for the ATT of the loaned group, if the households that have lifted out of poverty who have obtained microcredit for the poverty-alleviated population are not loaned, their production and operating income will decrease from 8.119 to 7.708, a decrease of 5.06%. For the ATU of the non-loaned group, if the households that have lifted out of poverty who have not obtained microcredit for the poverty-alleviated population are loaned, their production and operating income will increase from 7.851 to 9.815, an increase of 25.02%. From the perspective of monitoring objects, for the ATT of the loaned group, if the monitoring objects of obtaining microcredit for the poverty-alleviated population are not loaned, their production and operation income will decrease from 7.925 to 7.501, a decrease of 5.34%. For the ATU of the non-credited group, if the monitoring object of the non-credited microcredit of the poverty-alleviated population is loaned, its production and operation income will increase from 7.743 to 9.914, an increase of 28.04%. To sum up, the acquisition of microcredit for poverty-alleviated population by the households that have lifted out of poverty and the monitoring objects can significantly increase the income of production and operation, and the income increase effect of production and operation income of the monitoring object is higher than the households that have lifted out of poverty, which is 0.28% higher. This is mainly due to the continuous expansion of the coverage of the microcredit for poverty-alleviated population after 2020.Under the current policy framework of China, compared with the households who have lifted out of poverty, the monitoring object have become the key support object to prevent large-scale return to poverty at this stage. It is also the low-income group with the highest risk of returning to poverty in China. Local governments tend to focus more support resources on the monitoring object, so that

Table 7. Differences in the average treatment effect of different types of farmers.

| Households type | Group | Decision stage | | Average treatment effect | |
|---|---|---|---|---|---|
| | | Loaned | Unloaned | ATT | ATU |
| The households that have lifted out of poverty | Loaned group(N = 222) | 8.119 | 7.708 | 0.411*** | — |
| | | (0.03) | (0.037) | (0.047) | |
| | Unloaned group(N = 107) | 9.815 | 7.851 | — | 1.964*** |
| | | (0.0487) | (0.055) | | (0.074) |
| Monitoring object | Loaned group(N = 244) | 7.925 | 7.501 | 0.423*** | — |
| | | (0.025) | (0.034) | (0.042) | |
| | Unloaned group(N = 165) | 9.914 | 7.743 | — | 2.171*** |
| | | (0.04) | (0.046) | | (0.061) |

the microcredit for the poverty-alleviated population has a better effect on the income increase of the monitoring object than the households that have lifted out of poverty.

## 5.5  Action mechanism

The microcredit policy for the poverty-alleviated population will promote the increase of farmers ' capital factors, but in agricultural production and operation activities, capital factors cannot directly drive farmers to increase their income. Instead, with the increase of capital factors, farmers will increase the material capital investment in production and operation, including pesticides, fertilizers and land, so as to revitalize agricultural production and operation, thus increasing the income of production and operation. The material capital investment of production and operation can directly reflect the effect of farmers ' revitalization of production and operation, and the social capital can also indirectly reflect the situation of farmers ' access to production and operation information. Therefore, this study measures the material capital investment of production and operation by the expenditure of farmers engaged in production and operation activities, and measures the social capital by the expenditure of farmers 'favors and gifts. and the two are logarithmically processed to explore the mechanism of the microcredit policy for poverty-alleviated population to promote the increase of farmers 'income. As shown in Table 8, the estimated coefficient of the impact of farmers 'access to microcredit for the poverty-alleviated population on their physical capital investment in production and operation is 0.834, and it is significant at the statistical level of 1%.The estimated coefficient of the impact of farmers 'access to microcredit for the poverty- alleviated population on their social capital is 0.587, and it is significant at the statistical level of 5%. It can be concluded that the microcredit policy for the poverty- alleviated population promotes the increase of farmers ' income by promoting them to increase their physical capital investment and social capital investment. Hypothesis 2 was established.

## 5.6.  The analysis of difference in the effect of obtaining microcredit for the poverty-alleviated population on the production and operating income of farmers with different initial endowments

Considering that the production and operation income of farmers may be skewed or abnormal, the model estimation of the impact of microcredit for poverty-alleviated population on the production and operation income of farmers will be biased. At the same time, the production and operation income of farmers is used as the basis for measuring the initial endowment of farmers' production and operation, which can show the impact of microcredit for poverty-alleviated population on the production and operation income from the whole distribution of production and operation income. Based on this, this study uses the quantile regression model to estimate the difference in the effect of microcredit for poverty-alleviated population on the production and operation income of farmers with different initial endowments. Drawing on the ideas of Li Yajuan and Ma Ji [33], this study classifies farmers with production and operating income at 10%, 25%, 50%, 75%, and 90% quantiles as low, medium-low, medium, medium-high, and high initial endowment households, respectively.

As shown in the quantile regression results of Table 9, there is a significant difference in the impact of obtaining microcredit for poverty- alleviated populations on the production and operational income of farmers. At the 10%, 25%, 50% and

**Table 8.  The mechanism path of microcredit for poverty-alleviated population to increase farmers ' income.**

| Variable | Production and operation of material capital | Social capital |
|---|---|---|
| Microcredit for the poverty- alleviated population | 0.834*** | 0.587** |
|  | (0.192) | (0.239) |
| Control variable | control | control |
| R² | 0.132 | 0.028 |
| Sample size | 735 | 728 |

75% quantiles, the coefficients of the impact of obtaining microcredit for poverty- alleviated populations on the income of farmers' production and operation are 0.456, 0.372, 0.291, 0.159, respectively. The effect of increasing the income of production and operation income of farmers with low, medium and low initial endowments who obtain microcredit for poverty-alleviated population is greater than that of farmers with medium and high initial endowments. It is not difficult to see that the efficiency effect is decreasing as a whole(Fig 3), which can verify the research hypothesis 2. For small farmers, the credit funds invested in expanding the scale of production and operation have a marginal diminishing effect, so for the households that that have lifted out of poverty and monitoring objects with low initial endowments, the income-increasing effect of investment credit on farmers' production and operation income is greater than that of high initial endowments. That is to say, the microcredit for the poverty-alleviated population has a ' raising the low ' effect, and it also shows that the microcredit for the poverty-alleviated population has a relatively limited effect on the income increase of farmers with high initial endowments. In addition, at the 90% quantile, the effect of obtaining microcredit for poverty

**Table 9. Quantile regression of microcredit for poverty-alleviated population on farmers' production and operation income.**

| Variable | Explained variable: The natural logarithm of per capita production and operation income | | | | |
|---|---|---|---|---|---|
| | (1) | (2) | (3) | (4) | (5) |
| | q10 | q25 | q50 | q75 | q90 |
| Microcredit for poverty-alleviated population | 0.456*** | 0.372** | 0.291*** | 0.159* | 0.0880 |
| | (0.156) | (0.148) | (0.0995) | (0.0869) | (0.138) |
| control variable | control | control | control | control | control |
| constant term | 5.374*** | 6.331*** | 8.059*** | 9.264*** | 8.918*** |
| | (0.627) | (0.566) | (0.440) | (0.452) | (0.666) |
| $R^2$ | 0.1461 | 0.1202 | 0.112 | 0.1013 | 0.0797 |
| sample size | 738 | 738 | 738 | 738 | 738 |

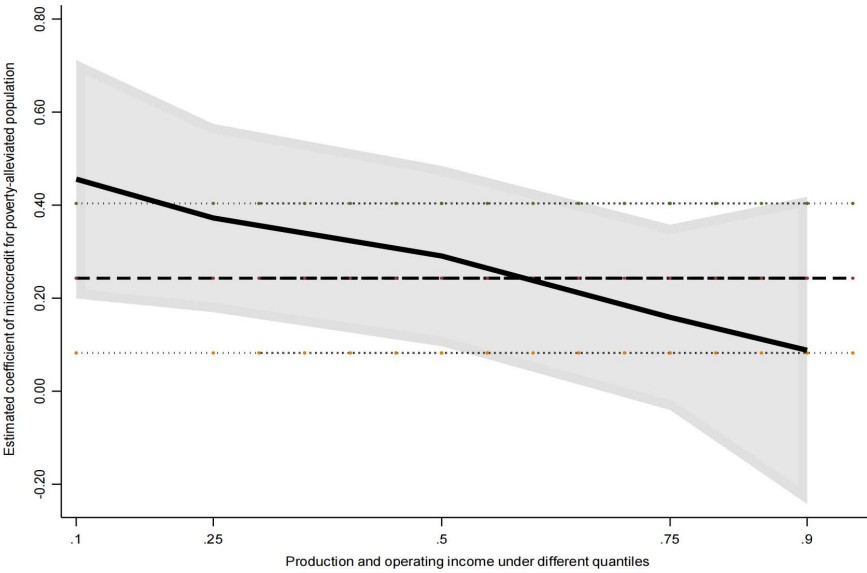

**Fig 3. Full quantile regression coefficient and its change.**

alleviation population on the income increase of production and operation income of farmers with high initial endowment is not significant, and the coefficient is much smaller than other quantiles. The possible reason is that the production and operation scale of high initial endowment households is relatively large, and the role of microcredit for the poverty-alleviated population with a cap of 50,000 in expanding the scale of production and operation is relatively limited. It can be seen that in order to promote the production and operation income of poverty-alleviated households and monitoring objects with high initial endowment, it is necessary to seek other paths other than the microcredit policy for the poverty-alleviated population.

Land is an important carrier for farmers to carry out production and operation activities, and it is also one of the main sources for farmers to obtain production and operation income. Therefore, according to the actual operating land area of farmers, this study divides the initial endowment situation into small-scale households with lower initial endowment of land (the actual operating area of land≤9) and large-scale households with higher initial endowment of land (the actual operating area of land> 9). The OLS model is used for grouping regression. The regression results are shown in Table 10. When farmers are small-scale households, the estimated coefficient of the impact of microcredit for poverty-alleviated population on their production and operating income is 0.293, and it is significant at the statistical level of 1%. It shows that for small-scale households with low initial endowment of land, their production and operation investment is relatively low, but the marginal income of production and operation investment is high. If they obtain credit funds, they can revitalize their production and operation activities under the existing production scale conditions and increase their income. When farmers are large-scale households, the impact of obtaining micro-credit for the poverty-alleviated population on their production and operation income is not significant and the coefficient is relatively small. There are two possible reasons. First, the production and operation investment of large-scale households is relatively large. In order to maintain or expand the scale of production and operation, financing through a limited amount of microcredit for the poverty-alleviated population may not meet the capital needs. Second, large-scale households are more likely to achieve economies of scale, and their production and operating income is relatively high, and they do not need to be financed by obtaining microcredit for poverty-alleviated population. This also verifies hypothesis 2, that is, microcredit for the poverty-alleviated population has a ' raising the low ' effect, and the acquisition of microcredit for the poverty-alleviated population has a higher income-increasing effect on farmers with low initial land endowment.

## 6. Conclusion and discussion

### 6.1. Conclusion

Based on the micro-survey data of 6 poverty-alleviated counties (cities) in Hotan Prefecture, Xinjiang Uygur Autonomous Region, covering the households that have lifted out of poverty and monitoring objects, this study uses the endogenous

**Table 10. Differences in the effect of microcredit on the income increase of poverty-alleviated population under different production scales.**

| Variable | (1) | (2) |
|---|---|---|
| | Small-scale households(The actual operating area of land ≤ 9) | Large-scale households (Actual operating area of land>9) |
| Microcredit for poverty -alleviated population | 0.293*** | 0.198 |
| | (0.0955) | (0.175) |
| Control variable | control | control |
| Constant term | 7.397*** | 8.573*** |
| | (0.409) | (0.750) |
| R2 | 0.153 | 0.244 |
| Sample size | 571 | 167 |

transformation model to investigate the impact of microcredit poverty-alleviated population on the production and operation income of farmers and its heterogeneity. At the same time, the quantile regression model and OLS regression model are used to explore the difference in the effect of microcredit for poverty-alleviated population on the production and operation income of farmers with different initial resource endowments. The research results show that, first, the microcredit of the poverty-alleviated population has an income-increasing effect. Specifically, under the counterfactual hypothesis, if the farmers who obtain the microcredit for the poverty-alleviated population are not loaned, their production and operation income will decrease by 3.31%, and if the farmers who do not obtain the microcredit for the poverty-alleviated population are loaned, their production and operation income will increase by 27.06%. After using the three robustness test methods of propensity score method (PSM), tail reduction and replacement of explanatory variables, the conclusion is still valid. In addition, both the households that have lifted out of poverty and the monitoring objects can significantly increase the income of production and operation, and the effect of income increase is 0.411 and 0.423 respectively, and the income-increasing effect of the monitoring objects is better than the households that have lifted out of poverty. Second, the estimated coefficient of the impact of farmers 'access to microcredit for the poverty-alleviated population on their physical capital investment in production and operation is 0.834, which is significant at the statistical level of 1%. The estimated coefficient of the impact on their social capital is 0.587, which is also significant at the statistical level of 1%. This shows that the microcredit policy for the poverty-alleviated population promotes income growth by encouraging farmers to increase their physical capital investment in production and operation and strengthen social capital. Third, the microcredit for the poverty-alleviated population has a 'raising the low 'effect. At the 10%, 25%, 50% and 75% quantiles, the coefficients of the impact of the microcredit of the poverty-stricken population on the income of farmers ' production and operation are 0.456,0.372,0.291 and 0.159, respectively, that is, the income-increasing effect of the microcredit of the poverty-alleviated population on farmers with low and medium-low initial endowments is greater than that of farmers with medium and medium-high initial endowments. and when farmers are small-scale households, the estimated coefficient of the impact of obtaining microcredit for the poverty-alleviated population on their production and operation income is 0.293, which is significant at the statistical level of 1%, while the coefficient of large-scale households is small and not significant, indicating that obtaining the microcredit for the poverty-alleviated population has a high income-increasing effect on farmers with low land initial endowments. On the whole, the synergistic effect of the micro-credit for the poverty-alleviated population on farmers with ' low→high ' initial resource endowments is decreasing as a whole, which also reflects that the microcredit for the poverty-alleviated population has a relatively limited income-increasing effect on farmers with high initial endowments.

## 6.2. Discussion

This paper empirically explores the income effect and heterogeneity of the microcredit policy of the poverty-alleviated population on the monitoring objects and the households that have lifted out of poverty during the period when China consolidates and expands the achievements of poverty alleviation and rural revitalization. The development of rural inclusive finance in China, especially the implementation of the microcredit policy for poverty alleviation, has enhanced the endogenous motivation of rural poor households, reduced the entry threshold for obtaining formal financial loans, alleviated financial exclusion [12], helped farmers solve the problem of lack of funds, obtained more initial funds for further production and operation investment activities, and improved the production and operation ability of poor households [14]. The research results of this paper are basically consistent with the above research conclusions. However, with the policy adjustment, the monitoring objects are included in the group enjoying the microcredit policy for the poverty-alleviated population, and the academic community has not clearly paid attention to whether the microcredit for the poverty-alleviated population can promote the income increase of the monitoring objects. This study further finds that the microcredit for the poverty-alleviated population not only has a significant income-increasing effect on the households that have lifted out of poverty, but also has a significant income-increasing effect on the monitoring objects, and the income-increasing effect of the microcredit for the poverty-alleviated population on the monitoring objects is greater than that of the households that

have lifted out of poverty. In terms of the role of the mechanism, this study finds that the micro-credit policy for the poverty-alleviated population promotes the increase of income by promoting the increase of material capital investment and social capital investment of farmers. Therefore, when promoting the microcredit policy for the poverty-alleviated population, we should also focus on the material capital investment and social capital investment of farmers, smooth the mechanism path, and improve the efficiency of policy implementation of microcredit for the poverty-alleviated population, so as to better increase income.In addition, there is controversy in the academic community about the internal differences in the income-increasing effect of China 's microcredit policy for poverty-alleviated population. Some scholars believe that the income-increasing effect on middle-income farmers is the best [20], and some scholars believe that the income-increasing effect on high-income farmers is the best [21]. Different from previous studies, from the perspective of research, this study not only considers the internal differences of income, but also considers the land resource endowment of farmers. In terms of research results, this study finds that the microcredit policy for the poverty-alleviated population has the best effect on the income increase of low-income farmers and small-scale farmers. Considering the initial resource endowment level of farmers, and the microcredit for poverty-alleviated population has a 'raising the low' effect, that is, The effect of microcredit for the poverty-alleviated population on the income increase of farmers with the ' low→high 'initial resource endowment is decreasing as a whole.When the initial resource endowment of the monitoring object or the households that have lifted out of poverty is high, the income increase effect of continuing to enjoy the microcredit policy for poverty-alleviated population is small. Therefore, it can be seen that the microcredit policy for the poverty-alleviated population should be more inclined to farmers with poor initial resource endowments. Since the policy has a limited effect on increasing the income of farmers with higher initial resource endowments, other credit policies should be further considered to increase the income of such farmers. At the same time, this paper also has the following limitations: First, the study area mainly selects typical representative areas, and the sample area is relatively small; second, the main use of cross-sectional data to carry out research, the lack of dynamic panel data support.

### 6.3. Policy implications

Based on the above research conclusions, this paper draws the following enlightenment: First, we should give full play to the important role of the microcredit policy for the poverty-alleviated population in consolidating the effective connection between the achievements of poverty alleviation and rural revitalization, enhance the " hematopoietic " function of precision inclusive finance, and strengthen the policy publicity and guidance for the households that have lifted out of poverty and monitoring objects who have the conditions to develop production and operation but have not obtained this loan, so as to achieve full credit; at the same time, strengthen supervision over the use of credit funds to ensure that households borrow and users return, and accurately use them for the development of production and operation. Second, increasing support for the production and operation of material capital investment of lenders, helping them purchase means of production, equipment and technology, improving production efficiency, and increase the promotion and publicity of agricultural technology training, give full play to the social capital of the lenders, which can indirectly enhance their ability to obtain information and resources, and promote the increase of production and operation income. Third, Strengthen the ' internal ' accuracy of the microcredit support policy for the poverty-alleviated population, In the long run, the microcredit policy for the poverty- alleviated population should be tilted to the monitoring objects with lower initial resource endowments and the households that have been lifted out of poverty. The inclusive credit policy for groups with high-income initial endowments should gradually consider 'gradual retreat ' or gradually shift to the inclusive commercial loan model.

### Supporting information

**S1 Data.**
(XLS)

## Author contributions

**Conceptualization:** Yin Liu, Lu Fan.

**Data curation:** Yin Liu.

**Formal analysis:** Lu Fan.

**Funding acquisition:** Yin Liu.

**Investigation:** Yin Liu.

**Methodology:** Lu Fan.

**Project administration:** Lu Fan, Binjian Yan.

**Resources:** Yin Liu, Lu Fan.

**Software:** Yin Liu.

**Supervision:** Binjian Yan.

**Visualization:** Yin Liu.

**Writing – original draft:** Yin Liu.

**Writing – review & editing:** Yin Liu, Lu Fan.

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
