## [Decision Letter · Decision Letter 0]

PONE-D-24-05024The Income-Increasing Effect of Microcredit for Poverty-Alleviated Population —Also on the "Raising the Low" EffectPLOS ONE

Dear Dr. Fan,

Thank you for submitting your manuscript to PLOS ONE. After careful consideration, we feel that it has merit but does not fully meet PLOS ONE’s publication criteria as it currently stands. Therefore, we invite you to submit a revised version of the manuscript that addresses the points raised during the review process.

Speicifically, pay attention to reorganizing the manuscript, improving language, clearly stating the objecives, providing more details in the methodology, and improving the policy recommendations section. 

We look forward to receiving your revised manuscript.

Kind regards,

Amar Razzaq, PhD

Academic Editor

PLOS ONE

Reviewers' comments:

Reviewer's Responses to Questions

**Comments to the Author**

1. Is the manuscript technically sound, and do the data support the conclusions?

Reviewer #1: Yes

Reviewer #2: No

2. Has the statistical analysis been performed appropriately and rigorously? 

Reviewer #1: Yes

Reviewer #2: I Don't Know

3. Have the authors made all data underlying the findings in their manuscript fully available?

Reviewer #1: No

Reviewer #2: No

4. Is the manuscript presented in an intelligible fashion and written in standard English?

Reviewer #1: Yes

Reviewer #2: No

5. Review Comments to the Author

Reviewer #1: This is a good piece of research focusing on much ignored aspect of poverty alleviation i.e. to focus already poverty alleviated segment of population. However, the manuscript would benefit from a thorough edit to improve the English. In addition, following are reservations:

1. Title is confusing, it should be rephrased to be attractive yet understandable.

2. Keywords are not given.

3. O page 4, objectives are not stated explicitly and clearly.

4. Sample size is reasonably good for this kind of survey, but readers do not seem to be given any information about reference year or research on behalf of which it was decided that population chosen NOW as 'poverty alleviated' was under poverty line ONCE.

5. The methodology of the research article is extravagant. It needs to make it concise!

6. There is no clear information about the counties and province (names should be given) chosen for collecting data and the reason/s to choose those particular counties.

7. For survey, “households with sudden severe difficulties, unstable poverty alleviation, and households with marginal poverty-prone households” were chosen but yardstick to differentiate these categories is not discussed. Authors need to add!

8. Authors are failed to describe what kind of different endowment conditions were used as benchmark to study the impact of microcredit on the production and operation income of farmers

9. Farmer's production and operation income, types of monitoring objects, should be defined at very start, where introduced in text first time to avoid reader to struggle with the concept.

10. Monitoring objects are defined too late to understand the previous context of study. It need be defined where this term is introduced first time in text.

11. Discussion of the results in the article is very scanty and needs re-do

12. Under 'Theoretical analysis and model construction' on page 7, authors mention precise implementation of microcredit policy. Before this recommendation, authors need to discuss current micro-credit policy in China, flaws in that policy and then they can suggest the precision in that policy.

13. Authors suggested policy makers to look at the initial resource endowments of debtor to get higher income effects, but they never showed that which initial level of resource endowment is considered to be effective and threshold.

14. The conclusions of the article should essentially capture the main findings of the research and the policy implication/recommendation. Conclusions by authors are too sumptuous!

15. In recommendations, “supervision of the use of credit funds, internal accuracy of the micro-credit assistance policy for the poverty-alleviated population, harmful consequences of overdue repayment, repayment of loans, and timely, appropriate renewal or extension of loans” were discussed which were not actually studied and analyzed. Recommendations should be in line with objectives and relevant results of study.

Reviewer #2: Comments to the authors

• Title is somewhat confusion. Poverty alleviated population?? What do you mean by it and clearly explain it in the beginning of introduction. I think title should be changed as well.

• Introduction should be rewritten and discuss the importance of the research topic, clear research gap, objectives and study contribution with adequate literature support not merely collection of words, sentences, and fluency.

• Review of literature section is missing.

• Manuscript needs rearrangement of section as well. Econometric methods should be placed after the data collection and hypothesis part.

• Sampling procedure is in-adequality explained. Authors should rewrite this whole section related to data collection and explain which sampling technique did they use in data collection. What was their sampling farm and what measures did they take to collect a representative sample. Moreover, authors should also clearly explain the reason behind selecting specific study region.

• Questionnaire used for cross sectional survey must be explained in a separate subsection. I think authors should have three clear subsection of study area selection, survey instrument, sampling procedure and data collection.

• Result section needs a better organization only study findings according to the study objective must be provided in this section. Supporting material should be in the in supplementary section or appendix.

• Discussion section should be separated from the results and literature support must be provided the study findings.

• Limitation of study is missing.

• Policy recommendation section is needed.

• More Literary support must be provided throughout the manuscript.

• Language of the manuscript is not good enough and must be improved.

6. PLOS authors have the option to publish the peer review history of their article (what does this mean? ). If published, this will include your full peer review and any attached files.

**Do you want your identity to be public for this peer review?** For information about this choice, including consent withdrawal, please see our Privacy Policy .

Reviewer #1: No

Reviewer #2: No

---

## [Author Response · Author response to Decision Letter 1]

5 Oct 2024

Thank you for the suggestions of reviewers and editors. At present, the expert review reply has been uploaded in the form of a file, the relevant data code has also been uploaded, and other content to be improved has been basically improved.

---

## [Decision Letter · Decision Letter 1]

PONE-D-24-05024R1Does Microcredit for the Poverty-Alleviated Population Have an

Income-Increasing Effect�—Also on the "Raising the Low" EffectPLOS ONE

Dear Dr. Yan,

Thank you for submitting your manuscript to PLOS ONE. After careful consideration, we feel that it has merit but does not fully meet PLOS ONE’s publication criteria as it currently stands. Therefore, we invite you to submit a revised version of the manuscript that addresses the points raised during the review process.

We look forward to receiving your revised manuscript.

Kind regards,

Amar Razzaq, PhD

Academic Editor

PLOS ONE

Journal Requirements:

Reviewers' comments:

Reviewer's Responses to Questions

**Comments to the Author**

1. If the authors have adequately addressed your comments raised in a previous round of review and you feel that this manuscript is now acceptable for publication, you may indicate that here to bypass the “Comments to the Author” section, enter your conflict of interest statement in the “Confidential to Editor” section, and submit your "Accept" recommendation.

Reviewer #1: All comments have been addressed

Reviewer #3: All comments have been addressed

Reviewer #4: All comments have been addressed

2. Is the manuscript technically sound, and do the data support the conclusions?

Reviewer #1: Yes

Reviewer #3: Yes

Reviewer #4: Yes

3. Has the statistical analysis been performed appropriately and rigorously? 

Reviewer #1: Yes

Reviewer #3: Yes

Reviewer #4: Yes

4. Have the authors made all data underlying the findings in their manuscript fully available?

Reviewer #1: Yes

Reviewer #3: No

Reviewer #4: No

5. Is the manuscript presented in an intelligible fashion and written in standard English?

Reviewer #1: Yes

Reviewer #3: Yes

Reviewer #4: Yes

6. Review Comments to the Author

Reviewer #1: (No Response)

Reviewer #3: Originality of the paper

The manuscript is generally intelligible and properly communicates the key ideas. The study presents findings based on original survey data collected from 738 farmers in six cities in Hotan Prefecture, Xinjiang. The study examines the "income-increasing effect" and "raising the low effect" of microcredit for the poverty-alleviated population in Xinjiang, China. This is a pertinent topic, particularly under the framework of China's rural revitalization strategy.

The dataset is novel and specific to the region, the originality. The theoretical model and hypotheses do not demonstrate significant innovation compared to prior studies cited in the literature (e.g page 4). The paper extensively builds on existing literature but does not convincingly establish how its theoretical framework and empirical models uniquely advance prior research.

It is therefore important to highlight/strengthen what differentiates this study from existing research and clarify how the findings contribute novel insights.

Literature Review

The literature review covers a wide range of methodologies and outcomes related to microcredit and poverty alleviation which sets the stage for justifying the empirical approach.

However, several key sources and alternative perspectives appear to be missing. For example, international literature on the effectiveness of microcredit in varied contexts could enrich the analysis or paper. Hence, consider incorporating more diverse viewpoints, including contrasting studies where microcredit has shown limited or no benefits and enhance the review's structure to ensure a more logical progression.

Methodology

The sampling procedures in the study are not fully transparent (e.g in page 10) i.e the justification for selecting six counties and the sampling strategy within villages require elaboration.

Moreover, data collection occurred in January 2023 but it lacks insights on recall biases that might affect responses (page 10). In this case it’s vital to provide more information on the sampling frame, data cleaning, and potential biases.

The statistical methods, including ESR and IV-2SLS, are appropriately chosen for addressing endogeneity and selection bias. In addition, the robustness checks (e.g., PSM, tailing treatment) are detailed.

However, the justification for the choice of instrumental variables should be elaborated further, especially their theoretical validity.

Additionally, the rationale for grouping households by age in instrumental variable construction is not fully explained (page 11-12).

Results Presentation and Interpretation

The paper demonstrates evidence of analytical rigor, with models addressing endogeneity and selection bias effectively being applied.

There is inconsistent phrasing in results interpretation, e.g., "income-increasing effect of obtaining microcredit" is sometimes vague and lacks contextual depth.

The statistical insights, such as counterfactual scenarios, are not accompanied by a clear narrative explaining their broader implications for policy or theory. Therefore, let the authors provide a narrative synthesis to bridge statistical findings with real-world applications and policies. It will also be interesting to clarify whether reported impacts align with hypothesized outcomes of the study.

The results are detailed, with statistical outputs well-organized into tables.

The use of probability density plots to visualize ATT and ATU enhances comprehension (page 23).

Some results lack adequate interpretation in terms of broader implications, e.g., why higher-income farmers exhibit smaller returns from microcredit (page 27-28).

Ethical Considerations

The ethical adherence is well stated, asserting that no approval was needed since the study involved survey data but there is no mention of consent processes, data anonymization, or data security for respondents. Thus it would be in order to provide explicit description of measures taken to ensure ethical compliance during data collection.

Data Availability

Data availability is declared as unrestricted however the instructions for accessing the dataset are unclear.

Conclusions & policy recommendations

Conclusions align with the data presented and confirm the hypotheses. However they overgeneralize findings, e.g., recommending universal microcredit policies. There is a need to tailor conclusions to specific findings, and include actionable policy recommendations based on household heterogeneity. In addition, the "raising the low" effect is not adequately expanded upon in the policy context.

Reviewer #4: General comments

1. The manuscript is technically sound, and the data support the conclusions.

2. The manuscript develops adequate robustness check models.

3. “Raising the Low" This need to come out clearly across the paper – from abstract to conclusion.

4. In all Tables, remove the vertical lines in all tables

1 Abstract

1.1 The policy recommendations are too broad, for instant, you recommend the need to promote the microcredit policy of the poverty-alleviated population from the aspects of policy stability and implementation precision. How and by doing what? (Page 1)

1.2 What policy can the study recommend for the for different categories of poverty-alleviated population.

2 Introduction

2.1 The paragraph is too long split it.

2.2 Discuss the novelty of the paper. The statement, “it is of great practical significance to study the income increase effect and raising the low effect of microcredit for the poverty-alleviated population” is not adequate. (Page 2 last sentence)

3 Literature review

3.1 In page 4, you refer to the period of poverty alleviation; which period is this? Has the period ended or is China still under the transition period of poverty alleviation?

3.2 So why is microcredit segmentation analysis for different categories of poverty-alleviated population important?

4 Descriptive results

4.1. I was expecting to see the descriptive result for different categories of poverty-alleviated population (Table 3)

5 Empirical Test and Result Analysis

5.1

While introducing 5.1.1 (Point out that it explains the selection / decision equation) and for the introductory part of 5.1.2 (Point out that it explains the result / outcome equation.

5.2 Though the results section is separated from the discussion, it is too long. The study needs to highlight major findings.

6 Conclusion and Discussion

6.1 Draw the conclusions must be drawn appropriately based on the data presented.

6.2 Conclusions need to be based on objectives and major findings and provide policy implications based on the findings.

6.3 Why is the discussion presented after conclusion (Section 6.2), this is confusing to the reader.

7 Reference

Format reference listed consistently according to PLOS ONE author guidelines.

The paper can be accepted with minor revisions.

7. PLOS authors have the option to publish the peer review history of their article (what does this mean? ). If published, this will include your full peer review and any attached files.

**Do you want your identity to be public for this peer review?** For information about this choice, including consent withdrawal, please see our Privacy Policy .

Reviewer #1: **Yes: ** Almazea Fatima

Reviewer #3: No

Reviewer #4: No

---

## [Author Response · Author response to Decision Letter 2]

21 Feb 2025

We have completed the revision of the paper.We will resubmit the revised paper to you, please review it again. If you have any other questions or need further discussion, you can contact us at any time. Thank you again for your review and guidance of our manuscripts.

---

## [Decision Letter · Decision Letter 2]

Does Microcredit for the Poverty-Alleviated Population Have an

Income-Increasing Effect�—Also on the "Raising the Low" Effect

PONE-D-24-05024R2

Dear Dr. Yan,

We’re pleased to inform you that your manuscript has been judged scientifically suitable for publication and will be formally accepted for publication once it meets all outstanding technical requirements.

Kind regards,

Amar Razzaq, PhD

Academic Editor

PLOS ONE

Additional Editor Comments (optional):

Reviewers' comments:

Reviewer's Responses to Questions

**Comments to the Author**

1. If the authors have adequately addressed your comments raised in a previous round of review and you feel that this manuscript is now acceptable for publication, you may indicate that here to bypass the “Comments to the Author” section, enter your conflict of interest statement in the “Confidential to Editor” section, and submit your "Accept" recommendation.

Reviewer #5: All comments have been addressed

2. Is the manuscript technically sound, and do the data support the conclusions?

Reviewer #5: Yes

3. Has the statistical analysis been performed appropriately and rigorously? 

Reviewer #5: Yes

4. Have the authors made all data underlying the findings in their manuscript fully available?

Reviewer #5: Yes

5. Is the manuscript presented in an intelligible fashion and written in standard English?

Reviewer #5: Yes

6. Review Comments to the Author

Reviewer #5: The author has made detailed revisions based on expert opinions. I think the manuscript has been thoroughly revised and can be recommended for acceptance.

7. PLOS authors have the option to publish the peer review history of their article (what does this mean? ). If published, this will include your full peer review and any attached files.

**Do you want your identity to be public for this peer review?** For information about this choice, including consent withdrawal, please see our Privacy Policy .

Reviewer #5: No

---

## [Editor Report · Acceptance letter]

PONE-D-24-05024R2

PLOS ONE

Dear Dr. Yan,

I'm pleased to inform you that your manuscript has been deemed suitable for publication in PLOS ONE. Congratulations! Your manuscript is now being handed over to our production team.

Kind regards,

on behalf of

Associate Professor Amar Razzaq

Academic Editor

PLOS ONE